# Pharmacokinomic Profiling Using Patient-Derived Cell Lines Predicts Sensitivity to Imatinib in Dermatofibrosarcoma Protuberans

**DOI:** 10.3390/cells14120884

**Published:** 2025-06-11

**Authors:** Rei Noguchi, Takuya Ono, Kazuki Sasaki, Mari Masuda, Akira Kawai, Yuki Yoshimatsu, Tadashi Kondo

**Affiliations:** 1Division of Rare Cancer Research, National Cancer Center Research Institute, 5-1-1 Tsukiji, Chuo-ku, Tokyo 104-0045, Japan; renoguch@ncc.go.jp (R.N.); takuono@ncc.go.jp (T.O.); 2Department of Oncopeptidomics, Tochigi Cancer Center, 4-9-13 Yohnan, Utsunomiya 320-0834, Japan; kasasaki@tochigi-cc.jp; 3Department of Peptidomics, Sasaki Institute, 2-2 Kanda-Surugadai, Chiyoda-ku, Tokyo 101-0062, Japan; 4Department of Proteomics, National Cancer Center Research Institute, 5-1-1 Tsukiji, Chuo-ku, Tokyo 104-0045, Japan; mamasuda@ncc.go.jp (M.M.); akawai@ncc.go.jp (A.K.); 5Department of Musculoskeletal Oncology and Rehabilitation Medicine, National Cancer Center Hospital, 5-1-1 Tsukiji, Chuo-ku, Tokyo 104-0045, Japan; 6Department of Patient-Derived Cancer Model, Tochigi Cancer Center, 4-9-13 Yohnan, Utsunomiya 320-0834, Japan

**Keywords:** dermatofibrosarcoma protuberans, sarcoma, patient-derived cell line, kinase activity, imatinib, drug screening, pharmacokinomics, biomarker

## Abstract

Dermatofibrosarcoma protuberans (DFSP) is a rare sarcoma, characterized by a *COL1A1-PDGFB* fusion. Imatinib, a multi-target tyrosine kinase inhibitor, is a standard treatment of DFSP. However, resistance emerges in 10–50% of cases. There is an urgent need for predictive biomarkers to refine the patient selection and improve therapeutic outcomes. We aimed to identify predictive biomarkers for imatinib response and explored a pharmacokinomic approach using in vitro assays with patient-derived DFSP cell lines. Four DFSP cell lines that we established were analyzed for tyrosine kinase activities on PamChip, a three-dimensional peptide array, in the presence and absence of imatinib, along with an imatinib-sensitive cell line, GIST-T1, as a positive control. Drug screening was also performed using 60 FDA-approved tyrosine kinase inhibitors, including imatinib. The kinomic profiles were compared with the kinase inhibitor screening results to identify predictive druggable targets. Drug sensitivity was associated with increased activity of PDGFRB, as indicated by the PamChip assay and Western blotting. Furthermore, imatinib sensitivity correlated with the activity of three kinases: FER, ITK, and VEGFR1, suggesting their potential as potential predictive biomarkers. Our cell-based pharmacokinomic approach using patient-derived DFSP cell lines would facilitate the identification of resistant cases to imatinib and guide alternative therapeutic strategies.

## 1. Introduction

Dermatofibrosarcoma protuberans (DFSP) is a rare mesenchymal tumor with an estimated incidence of 0.8 to 4.5 cases per million individuals per year [1,2]. DFSP is characterized as a superficial and locally aggressive fibroblastic neoplasm with a storiform cellular appearance. A defining molecular feature of DFSP is the presence of a collagen type I alpha 1-platelet-derived growth factor beta chain (*COL1A1-PDGFB*) fusion gene [1]. DFSP predominantly affects young to middle-aged adults, with the highest incidence occurring between 20 and 50 years of age, while pediatric DFSP accounts for approximately 6% of all cases [2,3]. DFSP is more common in males than in females, although the genetic factors underlying this difference remain unclear [4,5,6]. A study on DFSP patients reported that 42% of tumors occurred on the trunk, 23% on the upper extremities, 18% on the lower extremities, and 16% on the head and neck [1]. DFSP generally progresses slowly and carries a low risk of distant metastasis. However, due to its infiltrative growth pattern, local recurrence is common, with reported rates ranging from 10% to 60% [3,4]. Distant metastasis occurs in approximately 1% to 4% of cases. Additionally, fibrosarcomatous (FS) transformation is associated with tumor progression, an increased risk of metastasis, and higher proliferative activity. 

Imatinib, a PDGFR inhibitor, is approved for treating patients with unresectable or metastatic DFSP based on its molecular mechanism [5]. DFSP is driven by a specific chromosomal translocation t(17;22) [6], resulting in the fusion of the COL1A1 gene at chromosome band 17q21.33 and the PDGFB at chromosome band 22q13.1. This fusion leads to the production of a *COL1A1–PDGFB* fusion protein, which causes excessive PDGFB production and continuous activation of the PDGFB signaling pathway. However, a meta-analysis of nine studies, including 152 DFSP patients treated with imatinib, reported disease progression in 9.2% (14/152) of cases [7]. Additionally, the same studies revealed a mortality rate of 20% (21/105) among patients receiving imatinib [7]. Rutkowski et al. further reported that two independent phase II trials of imatinib in patients with locally advanced or metastatic DFSP demonstrated an objective response rate of approximately 50% and a one-year overall survival rate of 87.5% [8]. These findings emphasize the need for further investigation into the molecular mechanisms of imatinib resistance in DFSPs to develop biomarkers for predicting imatinib sensitivity.

Kinase activity profiling is essential for identifying predictive biomarkers of imatinib responsiveness in DFSPs. Protein phosphorylation is a fundamental mechanism that regulates cellular processes such as apoptosis, proliferation, migration, cell cycle, and differentiation [9]. The human genome encodes approximately 500 different kinases that orchestrate these critical functions [10], and nearly 90% of all proteins undergo phosphorylation [11]. Aberrant kinase activity, arising from genetic mutations such as amplification, point mutation, chromosomal translocation, and epigenetic modifications, contributes to carcinogenesis and cancer progression. While genomic and transcriptomic approaches have identified many driver kinases in human cancer, the molecular landscape remains incomplete, as protein kinases are regulated at multiple levels, including translation, stability, and post-translational modifications [12]. To overcome the limitations of genomics in predicting drug response, direct assessment of kinase activity is essential. Additionally, dysregulation of autophosphorylation and kinase-to-kinase regulatory interactions further contribute to abnormal kinase activity. These disruptions ultimately lead to various oncogenic effects, such as impaired cell functions, malignant transformation, and resistance to chemotherapy [12]. Consequently, dysregulated kinases and their substrate proteins serve as biomarkers influencing cancer treatment strategies [13,14]. 

In the present study, we explored a pharmacokinomic approach integrating drug sensitivity data with kinase activity profiles to identify specific kinase dependencies and mechanisms of sensitivity in DFSP tumor cells. This integrated methodology enables the identification of patients whose tumors exhibit kinase activity patterns predictive of a favorable response to imatinib, thereby optimizing treatment selection and improving clinical outcomes for DFSP.

## 2. Materials and Methods

### 2.1. Patient Backgrounds

Clinical specimens used in this study were approved by the Institutional Review Boards of the National Cancer Center (NCC-IRB #2004-050). We analyzed four DFSP cell lines derived from individual patients with DFSP. These tumors were obtained from both primary and metastatic sites across different anatomical locations. Detailedclinical information is summarized in Table 1. 

### 2.2. Cell Culture Procedure

This study used four DFSP cell lines that were previously established in our lab, NCC-DFSP1-C1 [15], NCC-DFSP3-C1 [16], NCC-DFSP4-C1 [17], NCC-DFSP5-C1 [18], and a GIST cell line, GIST-T1. DFSP cells were cultured in DMEM/F-12 medium (Thermo Fisher Scientific Inc., Waltham, MA, USA) enriched with GlutaMAX (Thermo Fisher Scientific Inc.) and various supplements, including 5% heat-inactivated fetal bovine serum (FBS) (Thermo Fisher Scientific Inc.), 10 µM Y-27632 (ROCK inhibitor; Selleck Chemicals, Houston, TX, USA), 10 ng/mL basic fibroblast growth factor (bFGF; Sigma-Aldrich Co. LLC, St. Louis, MO, USA), 5 ng/mL epidermal growth factor (EGF; Sigma-Aldrich Co. LLC), 5 µg/mL insulin (Sigma-Aldrich Co. LLC), 0.4 µg/mL hydrocortisone (Sigma-Aldrich Co. LLC), 100 μg/mL penicillin, and 100 µg/mL streptomycin (Nacalai Tesque Inc., Kyoto, Japan). GIST-T1, derived from a pleural metastasis of gastric GIST in a Japanese woman, was obtained from CosmoBio (CosmoBio, Tokyo, Japan) and cultured in GISTM (CosmoBio). These cell lines were plated on a 10 cm collagen-coated culture dish (Fujifilm Co. Ltd., Tokyo, Japan) and maintained at 37 °C in a humidified incubator containing 5% CO_2_. To verify identity and ensure quality control, cell lines were routinely checked for mycoplasma contamination with an e-Myco *Mycoplasma* PCR Detection Kit (Intron Biotechnology, Gyeonggi-do, Korea) and subjected to short tandem repeat (STR) using the GenePrint 10 system (Promega Co., Madison, WI, USA) and a 3500xL Genetic Analyzer (Thermo Fisher Scientific Inc.).

### 2.3. Drug Screening for Tyrosine Kinase Inhibitors

Anti-proliferative effects of 60 tyrosine kinase inhibitors were measured, as previously described [19]. In brief, four DFSP cell lines and the GIST-T1 cell line were seeded in a 384-well plate (Thermo Fisher Scientific, Fair Lawn, NJ, USA) at a density of 5 × 10^3^ cells/well in DMEM/F12 medium supplemented with GlutaMAX (both Thermo Fisher Scientific Inc.), 5% heat-inactivated fetal bovine serum (FBS) (Gibco, Grand Island, NY, USA), and a cocktail of additives, including 10 µM Y-27632 (ROCK inhibitor; Selleck Chemicals), 10 ng/mL bFGF, 5 ng/mL EGF, 5 µg/mL insulin, 0.4 µg/mL hydrocortisone (all Sigma, St Louis, MO, USA), 100 μg/mL penicillin, as well as 100 µg/mL streptomycin (Nacalai Tesque). The GIST-T1 cell line was plated at a density of 5 × 10^3^ cells/well in a 384-well plate (Thermo Fisher Scientific Inc.) in the GISTM medium. Automated dispensing of all cell lines was performed with the Bravo automated liquid handling (Agilent Technologies, Santa Clara, CA, USA). The next day, a panel of 60 kinase inhibitors was added at 10 µM using the same system (Appendix A). After 72 h, the anti-proliferative effects of 60 kinase inhibitors were measured with the CCK-8 reagent (Dojin-do, Kumamoto, Japan). Cell viability was calculated as the percentage of growth inhibition relative to the DMSO control. The half-maximal inhibitory concentrations (IC_50_) were calculated. Each drug was identified in the initial screening study and for standard treatment drugs. The drugs were tested at 10 serial concentrations ranging from 0.1 nM to 100 µM using uniformly seeded cells. IC_50_ values were generated using GraphPad Prism 9.1.1 (GraphPad Software, San Diego, CA, USA).

### 2.4. Sample Preparation

DFSP PDCs and GIST cell lines were seeded into 6 cm dishes. The following day after seeding, imatinib was added at concentrations of 0.1, 1, and 10 µM. DMSO was used for the control. After 72 h, the medium was discarded, and the cells were washed twice with PBS and scraped off the plate using a cell scraper on ice. The detached cells were collected into 1.5 mL tubes, centrifuged, and pelletized. The supernatant was removed, and the cell pellets were collected.

### 2.5. Protein Extraction from Imatinib-Treated Cells

Protein extraction was performed, as previously described [20,21]. The cell pellets of four DFSP cell lines and GIST-T1 cell line were incubated with M-PER Mammalian Extraction Buffer (Pierce, Rockford, IL, USA) with Halt protease and phosphatase inhibitors (Pierce cat. 78420, 78415) on ice for 30 min. After centrifugation at 4 °C, 1500 rpm for 30 min, the supernatant was recovered as protein samples. The concentration of the extracted proteins was measured using Bradford reaction assay (Bio-Rad Laboratories, Inc., Hercules, CA, USA) with Microplate Spectrophotometer Epoch (BioTek Instruments, Vermont, WI, USA), and stored at −80 °C until use. 

### 2.6. In Vitro Tyrosine Kinase Activity Assay

A total of 5 μg of extracted protein was combined with m-PER buffer, ATP, and fluorescently tagged anti-PY20 antibodies. This mixture was then applied to the tyrosine (PTK) PamChips and analyzed using the PamStation 12 kinomics workstation (PamGene International, BJ’s-Hertogenbosch, The Netherlands) according to the PTK PamChip protocol in Evolve12 Software (v. 1.5), as previously described [22]. Protein lysates were perfused through the chip array, during which real-time imaging was performed. Signal acquisition and quantification were carried out using BioNavigator v. 5.1 (PamGene International). All experiments were run in duplicate for reproducibility. To assess global kinase activity, data were further analyzed using BioNavigator software v. 6.3.67.0 (PamGene International). The kinase activity-dependent signal intensities were recorded at each peptide ‘spot’ on the array. Fluorescence intensity from 196 sites was measured over sequential 50 ms exposure intervals during lysate perfusion. Additional signal measurement was taken at multiple exposure times (10, 20, 50, 100, 200 ms) following lysate removal. Intensity values were converted into slope values relative to the exposure time, scaled by 100, and subjected to log2 transformation for downstream analysis. 

### 2.7. Kinase–Substrate Enrichment Analysis (KSEA)

To estimate the relative activity of individual kinases, kinase–substrate enrichment analysis (KSEA) was applied to relative kinase activity signal data from the five cell lines. This approach relies on the principle that changes in the activity of a kinase are reflected by coordinated alterations in the phosphorylation levels of its downstream substrates. The analysis was performed using the KSEA application implemented in the R package (v.0.99.0) [23], based on a previously described algorithm [24]. Log2FC ratios of phosphopeptide signals served as input. KSEA was run for three different comparisons: 0.1 µM imatinib/DMSO, 1 μM imatinib/DMSO, and 10 μM imatinib/DMSO. Z-Scores were used as output for visualization. 

### 2.8. Gene Ontology Analysis

Gene Ontology (GO) analysis of kinase activity was performed using the R-package “ClusterProfiler” version 4.12.6. GO and the Kyoto Gene and Genomic Encyclopedia (KEGG) were analyzed. Raw *p*-values were calculated and adjusted using the false discovery rate (FDR) correction. Statistically, significance was defined as both *p*-value and FDR < 0.05. 

### 2.9. Correlation Analysis to Predict Drug Sensitivity for Imatinib

Spearman’s correlation method was applied, and *p*-values were adjusted using FDR correction. Differences were determined to be statistically significant when the *p*-value was less than 0.05. Calculations of Spearman correlations, *p* values, and visualization were performed by R software version 4.4.1. 

### 2.10. Western Blotting

The cell lysates used for in vitro kinase activity assay were lysed with 1x NuPAGE LDS sample buffer (Thermo Fisher Scientific Inc.) containing 50 mM dithiothreitol (DTT) and heated to 70 ℃ for 10 min. Samples (10 μg protein per lane) were subjected to NuPAGE bis-Tris 4 to 12% gel electrophoresis with a morpholinepropanesulfonic acid/sodium dodecyl sulfate (MOPS-SDS) running buffer (ThermoFisher Scientific Inc.). The proteins were transferred to 0.22-μm Immobilon-P transfer membranes (Millipore Corporation, Bedford, MA, USA). The membranes were incubated with the following primary antibodies at 4 ℃ overnight: rabbit anti-PDGFRB (#3169, Cell Signaling Technology, 1:1000, for detection of human PDGFRB), rabbit anti-phosphor-PDGFRB (#3124, Cell Signaling Technology, 1:1000, for detection of human PDGFRB phospho-Y1009), and mouse anti-GAPDH (MAB374, Merk, 1:1000, for detection of human GAPDH). Then the membranes were incubated with appropriate anti-mouse or anti-rabbit horseradish peroxidase-linked secondary antibodies (Anti-mouse: #7076P2, Cell Signaling Technology, 1:5000; and anti-rabbit: #7074P2, Cell Signaling Technology, 1:5000) at room temperature for 1h. The membranes were treated with Western Lightning Plus-ECL Kit (GE Healthcare, Little Chalfont, UK) to detect proteins. Images were acquired with ImageQuant LAS4000 mini (GE Healthcare) and analyzed by ImageJ 1.54d software.

## 3. Results

### 3.1. Patient Characteristics

Patient characteristics are summarized in Table 1. The median age was 51 years (range, 46–60 years). The study included three men and one woman. A COL1A-PDGFB fusion, a characteristic genetic feature of DFSP, was identified in all cases as previously reported [15,16,17,18]. Among the four DFSP PDCs (patient-derived cells), three were derived from primary tumor sites, while one originated from a metastatic site. Regarding tumor location, the four cases arose from distinct anatomical sites: the abdominal wall left thigh, left lower back, and left proximal tibia. In terms of diagnosis, three cases were classified as FS-DFSP, and one was categorized as DFSP. 

### 3.2. Sensitivity to Kinase Inhibitors

The anti-proliferative effects of 60 FDA-approved kinase inhibitors (Appendix A) were examined in four DFSP cell lines and a GIST-T1 cell line using a drug screening assay. DFSP cell lines generally exhibited resistance to these inhibitors, whereas the GIST-T1 cell line demonstrated sensitivity. Several inhibitors exhibited broad tumor-suppressive effects across all cell lines (Figure 1, Table 2, Appendix A). GIST-T1 was sensitive to imatinib and imatinib mesylate but the four DFSP cell lines were relatively unresponsive. Notably, eight kinase inhibitors consistently reduced cell growth (Figure 1, Table 2, Appendix A). These eight kinase inhibitors, listed in Figure 1, included cediranib, crizotinib, ceritinib, entrectinib, sunitinib malate, foretinib, osimertinib, and ponatinib (Table 2). 

### 3.3. Effect of Imatinib on Kinase Activity in Four DFSP Cell Lines and GIST-T1 Cell

The kinase activity profiles of DFSP PDCs and GIST-T1 cells were examined using a three-dimensional peptide array called “PamChip”. The PamChip is capable of assessing kinase activity for 98 tyrosine kinases with a small sample volume. Kinase activity profile data were successfully obtained for all cell lines. After quality control, 142 out of the 196 PTK peptides were evaluated. The mean signal intensity per peptide was calculated and log-2 was transformed (Appendix A). For all samples, the kinase activity assay was performed in duplicate. The results were visualized as a heatmap (Appendix A), where the rows represent individual peptides, and the columns represent each sample.

To elucidate changes in kinase activity under imatinib treatment, we conducted an exploratory inhibitor study using the three-dimensional peptide array in the four DFSP PDCs and imatinib-sensitive GIST-T1 cells. To ensure accuracy, fold changes were calculated as the ratio of the signal intensity in imatinib-treated samples to untreated samples for all five cell lines (Figure 2A, Appendix A). In total, fold change values were obtained from 142 peptide substrates. A fold change greater than two was defined as an increase, while a fold change less than 0.5 was defined as a decrease. We also plotted the fold changes to illustrate the differential effects of imatinib on each cell line (Figure 2B). Using this threshold, kinase activity on 10 µM imatinib treatment was decreased in 6 substrates (4.2%) in NCC-DFSP1-C1, 9 substrates (6.3%) in NCC-DFSP3-C1, 7 substrates (4.9%) in NCC-DFSP4-C1, 3 substrates (2.1%) in NCC-DGFP5-C1, and 52 substrates (36.6%) in GIST-T1. Conversely, kinase activity increased in eight substrates (5.6%) in NCC-DFSP1-C1, two substrates (1.4%) in NCC-DFSP3-C1, five substrates (3.5%) in NCC-DFSP4-C1, eight substrates (5.6%) in NCC-DGFP5-C1, and nine substrates (6.3%) in GIST-T1. 

### 3.4. Dynamic Change in Kinases by Imatinib

Using our kinase activity datasets, we inferred the upstream kinase activity using KSEA [24]. KSEA systematically infers global kinase pathway activation from phosphoproteomics experiments. The 142 identified phosphorylated peptide sites obtained from PamChip data were matched to kinase–substrate relationship databases. We used fold change from the 142 peptide sites. KSEA scores were calculated for four DFSP PDCs and GIST-T1 cells. The analysis identified the activity of 44 different kinases and phosphatases (Appendix A). Subsequently, KSEA scores were calculated, and hierarchical clustering was applied to visualize the 17 most significantly activated and inhibited kinases and phosphatases (Figure 3). As a result of KSEA, PDGFRB was significantly inhibited at all concentrations in GIST-T1 cells. Additionally, NCC-DFSP3-C1 cells showed significant inhibition of RET. Conversely, KSEA revealed significant activation of RET on all concentrations in NCC-DFSP1-C1 cells, along with activation trends for INSR and PTK6 in NCC-DFSP4-C1 and NCC-DFSP5-C1 cells (Figure 3). 

We analyzed the protein expression of significantly regulated kinases to assess whether the comparative analyses of kinomic profiles between sensitive and resistant cell lines were reflected in changes in phosphorylation levels of PDGFRB at tyrosine 1009 using Western blotting (Appendix A). An upregulation of phosphorylated PDGFRB (p-PDGFRB) was observed in NCC-DFSP4-C1 cells, which were resistant to imatinib, while a downregulation of p-PDGFRB was observed in GIST-T1 and NCC-DFSP3-C1 cells as sensitive cell lines. These findings indicate a potential correlation between PDGFRB activation and sensitivity to imatinib in these cell lines.

### 3.5. Gene Ontology Analysis

Gene ontology (GO) analysis was performed using KSEA scores of the 142 peptides under treatment with 10 µM imatinib for each kinase to identify significant biological pathways associated with imatinib sensitivity. The GO analysis revealed multiple pathways in four DFSP PDCs and GIST-T1 cells (Table 3). The identified pathways are highly significant, with the *p*-values less than 0.05. The MAPK signaling pathway was upregulated in NCC-DFSP3-C1 and GIST-T1 cells, both of which are imatinib-sensitive. In addition, the EGFR tyrosine kinase inhibitor resistance pathway was significantly enriched and up-regulated in NCC-DFSP3-C1 cells. Conversely, in NCC-DFSP1-C1 cells, which are resistant to imatinib, the MAPK signaling pathway was significantly downregulated. In the remaining resistant cells, including NCC-DFSP4-C1 and NCC-DFSP5-C1 cells, no pathways showed significant enrichment.

### 3.6. Correlation Analysis of Drug Sensitivity on Imatinib and Kinase Activity

To determine whether kinase activity could serve as a predictive biomarker for imatinib response, we applied Spearman correlation analysis to kinomic profiling array data targeting kinases and imatinib drug sensitivity data (Figure 4A). Four significantly correlated target kinases were selected for imatinib: FER, FLT1, ITK, and PDGFRB, and their R values were determined by linear regression (Spearman correlation r > 0.95, *p* < 0.05) (Appendix A, Figure 4B).

## 4. Discussion

We employed a pharmacokinomic approach with DFSP PDCs to identify biomarkers predictive of imatinib sensitivity in patients. This is the first study to integrate kinase activity profiles with drug screening data from patient-derived cell lines under the pharmacokinomic concept. Our approach revealed that four kinases are associated with the imatinib response. Consequently, kinase activity profiling enabled the exclusion of individuals resistant to treatment and the selection of those likely to benefit. This pharmacokinomic approach will further support the stratification of DFSP patients and the preparation for other therapeutic strategies.

### 4.1. Patient-Derived DFSP Cell Lines

The strength of this study lies in the pharmacokinomic approach utilizing patient-derived DFSP cell lines. A drug discovery strategy with substantial evidence is required to develop effective treatments for DFSPs. Even though the use of cell lines has provided valuable information on various cancers, DFSP cell lines remain scarce. According to Cellosaurus (https://www.cellosaurus.org/ (accessed on 1 April 2025)) [25], a worldwide knowledge resource of cell lines, five cell lines of DFSP have been reported, four of which appear in the scientific literature [26,27,28]. Among the five DFSP cell lines, only one is currently available from a public cell bank, likely due to the limited availability of clinical specimens, the rarity of DFSP, and its indolent nature (Appendix A). To address this issue, we established five new DFSP cell lines [15,16,17,18]. Previous research by Chan JY et al. successfully generated the MDFSP-S1 cell line, derived from a 47-year-old Chinese female patient with imatinib-resistant DFSP with FS [28]. These cells were established from a recurrent biopsy sample following imatinib treatment. The study also examined drug sensitivities against 11 tyrosine kinase inhibitors, including osimertinib, gefitinib, imatinib, sunitinib, regorafenib, pazopanib, erlotinib, axitinib, palbociclib, everolimus, and olaparib in MDFSP-P1 cells. MDFSP-S1 cells demonstrated no significant response to imatinib, palbociclib, everolimus, olaparib, gefetinib, and erlotinib (IC_50_ all > 10 μM). However, moderate anti-proliferative effects were observed for antiangiogenic agents such as sunitinib, regorafenib, pazopanib, and axitinib and osimertinib, with IC_50_ values between 1 and 10 μM at 72 h. In our study, four DFSP PDCs were established from patients who had not received imatinib, and three DFSP cell lines were derived from primary tumors. Among the four DFSP cell lines, three accompanied FS changes. The three kinase inhibitors tested on MDFSP-P1, palbociclib, everolimus, and olaparib were not examined as we focused on tyrosine kinase inhibitors. Excluding these three agents, osimertinib and sunitinib were identified as highly suppressive groups with IC_50_ values ranging from 0.15 to 2.62 µM for osimertinib, and 4.68 to 9.41 µM for sunibinib. Additionally, the five tyrosine kinases inhibitors—ponatinib, foretinib, cediranib, crizotinib, and entrectinib—demonstrated growth-suppressing activity on our four DFSP cell lines, though they were not examined in MDFSP-S1. Comparing both cell line studies, imatinib exhibited no significant anti-tumor effects, irrespective of prior exposure. Moreover, the efficiency of imatinib was consistently low, regardless of the presence of FS findings. Further studies will be conducted to assess imatinib’s therapeutic value and identify alternative treatments.

### 4.2. Kinase Activity Profiles Under Imatinib Treatment on DFSP Cell Lines and GIST-T1 Cells

Our study revealed that in one of the four DFSP PDCs, RET was inhibited by imatinib in a dose-dependent manner, while GIST-T1 cells demonstrated PDGFRB inhibition. Imatinib is the first receptor tyrosine kinase (RTK) inhibitor with activity against ABL, BCR-ABL, PDGFR, and c-KIT [29], and is the standard of care in chronic myelogenous leukemia (CML), GIST, DFSP, and various other cancers [29]. The observed PDGFRB inhibition in GIST-T1 cells was consistent with target inhibition of PDGFR by imatinib, as well as its low IC_50_ value of 0.0018 µM. In contrast, the four DFSP PDCs exhibited high IC_50_ values, ranging from 4.4 to 32.7 µM. Among them, NCC-DFSP3-C1 cells had the lowest IC_50_ value (4.4 µM) and showed RET inhibition in a dose-dependent manner, whereas the other three PDCs did not exhibit significant inhibition. Two research groups, including Cohen MS et al. and De Groot JW et al., reported that imatinib inhibits both cell proliferation and RET phosphorylation in medullary thyroid carcinoma (MTC), a malignancy of C Cells of the thyroid gland associated with gain-of-function mutations in *RET* [30,31]. Their findings support the observed RET inhibition in the NCC-DFSP3-C1 cell line. De Groot JW et al. further suggested that RET tyrosine kinase can be inhibited by imatinib because both c-kit and RET belong to the same subfamily of tyrosine kinase receptors [30,31]. Although these studies focused on MTC rather than DFSP, their findings may provide indirect support for our findings. It has been well known that cells transformed with the *COL1A1-PDGFRB* gene, as well as cell cultures derived from DFSP patients, are inhibited by imatinib [32,33]. Furthermore, imatinib has been approved for clinical use in DFSP. However, previous in vitro studies have only assessed a limited portion of PDGFR using Western blotting, and no comprehensive evaluation of kinase activity or the mechanism by which imatinib acts on DFSP has been reported. Therefore, our report is the first to clarify the mechanism underlying the efficacy of imatinib in DFSP PDCs. Currently, the only known imatinib-resistant DFSP PDC listed in the Cellosaurus database is “MDFSP-C1”, making it impossible to compare imatinib-sensitive and resistant DFSP PDCs. To elucidate the mechanism of imatinib susceptibility, additional DFSP PDCs, including susceptible strains, need to be established. Moreover, a comprehensive evaluation of kinase activity using our in vitro kinase activity assay and phosphoproteomics is essential for deepening the understanding of the imatinib effect in DFSP.

### 4.3. Predictive Biomarkers of Imatinib Sensitivity

We identified four kinases FER, FLT1, ITK, and PDGFRB as predictive biomarkers of imatinib responsiveness based on our pharmacokinomics approach integrating in vitro kinase activity assay and drug sensitivity profiles. Among the four kinases, PDGFRB is a known target of imatinib and underlies its therapeutic activity in DFSP [34]. Several studies have investigated PDGFRB as a predictive biomarker of imatinib in DFSP. For example, Ugurel S et al. reported that weak PDGFRB phosphorylation was associated with non-response to imatinib. They also found that PDGFRB, EGFR, and insulin receptors were activated in a high percentage of DFSP samples using an RTK phosphorylation array assay on cryopreserved tumor tissues collected before and after imatinib treatment from seven patients [34]. However, no significant changes in PDGFRB phosphorylation levels were observed during treatment, and notably, no decrease in PDGFRB activation was seen even in responders [34]. In contrast, McArthur GA et al. reported dramatic clinical responses in DFSP patients whose tumors expressed relatively low levels of activated PDGFRB, as determined by Western blot analysis of frozen tumor specimens collected before treatment [35]. Similarly, Sjoblom T et al. identified the phosphorylation of PDGFRB in three DFSP PDCs and their in vivo models using immunoprecipitation assays [33]. They demonstrated that imatinib exerted growth-inhibitory effects both in vitro and in vivo through tumor cell apoptosis. In any case, since the sample collection and evaluation methods were different and the number of samples was small, it is difficult to reach conclusions. We propose that the rate of change in kinase activity before and after imatinib administration correlates with drug sensitivity. In the future, it will be necessary to increase the number of samples and examine whether the changes after imatinib administration are consistent.

FLT1, also known as vascular endothelial growth factor receptor 1 (VEGFR1), is upregulated in various cancers and has been associated with disease progression, poor prognosis, metastasis, and recurrent disease in humans [36]. Our study revealed that VEGFR1 was one of the predictive biomarkers for imatinib, based on a comparison between comprehensive kinase activity profiles and drug sensitivity to imatinib. Although VEGFR1 has not previously been reported as a biomarker for imatinib in DFSP, Mathew P et al. noted that plasma levels of VEGFR1 link with imatinib therapy in prostate cancer [37], which may support our findings. To the best of our knowledge, the remaining kinases—FER and ITK—were newly identified as predictive biomarkers for imatinib sensitivity in our study, which have not been previously reported in DFSP. In the present study, we were unable to conduct validations using clinical samples. If clinical samples or a greater number of preclinical models are available, the predictive roles of these kinases can be further confirmed. Altogether, these observations strongly support the utility of our pharmacokinomic profiling approach in identifying predictive biomarkers for imatinib and warrant further validation studies using additional samples.

### 4.4. Limitations

Our study has several limitations. First, we employed only four DFSP PDCs. In oncology research, findings obtained with a small number of models may not provide conclusive results. Therefore, it is essential to validate our observations using a broader panel of cell lines derived from different DFSP subtypes, such as DFSP and FS-DFSPs. Given the rarity of DFSP cases, collaborative efforts across multiple institutions are necessary to develop additional PDC models representing diverse forms of the disease. Second, clinical data regarding imatinib responsiveness in the patients from whom the four DFSP PDCs were established were not available, as not all individuals received the drug. Acquiring such information would allow us to assess whether the observed in vitro sensitivity correlates with therapeutic outcomes in clinical settings, thereby reinforcing the validity of our findings. Third, we used the GIST-T1 cell line as a representative imatinib-sensitive model because no imatinib-sensitive DFSP PDCs were available at the start of our study. Although the GIST-T1 cell line was derived from gastrointestinal stromal tumors and differs pathologically from DFSP, it served as a useful comparative reference. Future studies would benefit from using imatinib-sensitive DFSP cell lines, which would enable a more disease-relevant comparison and potentially yield more precise insights. Fourth, our findings are derived solely from in vitro data and lack validation in clinical samples or in vivo models. To confirm the clinical relevance of the identified biomarkers, further studies using patient-derived tumor samples or animal models will be required. Given the rarity of DFSP, such validation efforts will likely necessitate multiple institutional collaborations. Fifth, our findings of the mechanisms underlying resistance, based on in vitro kinase activity assay, remain speculative. To validate these hypotheses, additional functional studies, including overexpression and/or knockdown experiments, are necessary to strengthen the evidence. We plan to pursue such experiments in future research.

## 5. Conclusions

An integrated pharmacokinomic approach using patient-derived DFSP cell lines identified key mechanisms underlying drug sensitivity and predictive biomarkers to imatinib response. A common feature associated with the sensitivity was increased PDGFRB activity observed in four DFSP PDCs and GIST-T1 cells. Moreover, imatinib sensitivity correlated with the activity of three kinases: FER, ITK, and VEGFR1, indicating their potential as predictive biomarkers. In conclusion, our pharmacokinomic approach facilitates the efficient identification of drug sensitivity mechanisms and predictive biomarkers, paving the way for the development of novel therapeutic strategies involving imatinib for DFSPs.

## Figures and Tables

**Figure 1 cells-14-00884-f001:**
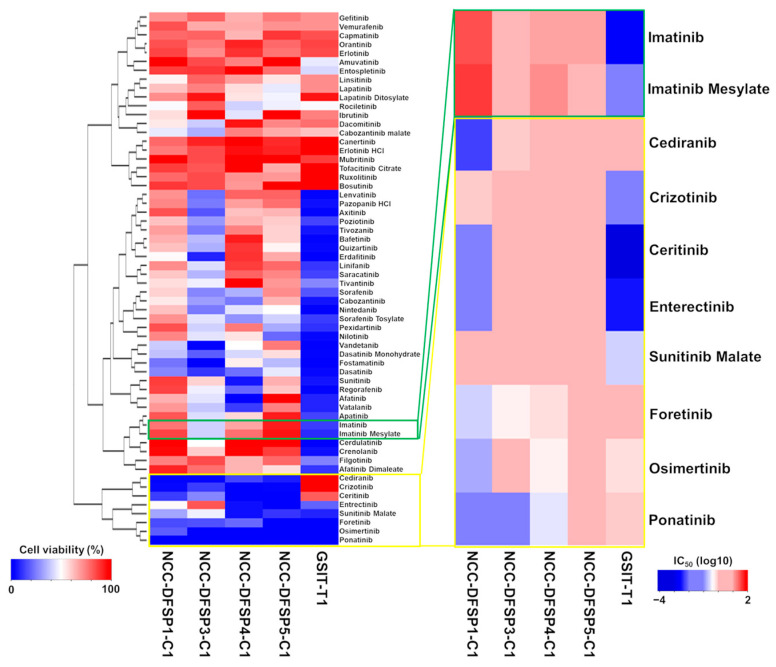
Summary of growth suppression data on 60 tyrosine kinase inhibitors in a panel of four DFSP cell lines and the GIST-T1 cell line. Left heatmap of anti-tumor suppressive responses of the cell lines to 60 tyrosine kinase inhibitors (10 µM). A low cell viability percentage (blue) indicates reduced cell growth, whereas a high cell viability percentage (red) indicates resistance. The yellow open box highlights imatinib and imatinib mesylate, and the green open box highlights the top eight kinase inhibitors that suppressed the cell growth in all five cell lines most strongly among the 60 kinase inhibitors. Right heatmap of IC_50_ values of the five cell lines to ten kinase inhibitors, including imatinib, imatinib mesylate, cediranib, crizotinib, certinib, entrectinib, sunitinib malate, foretinib, osimertinib, and ponatinib. Low IC_50_ values (blue) indicate reduced cell growth, whereas high IC_50_ values (red) indicate resistance.

**Figure 2 cells-14-00884-f002:**
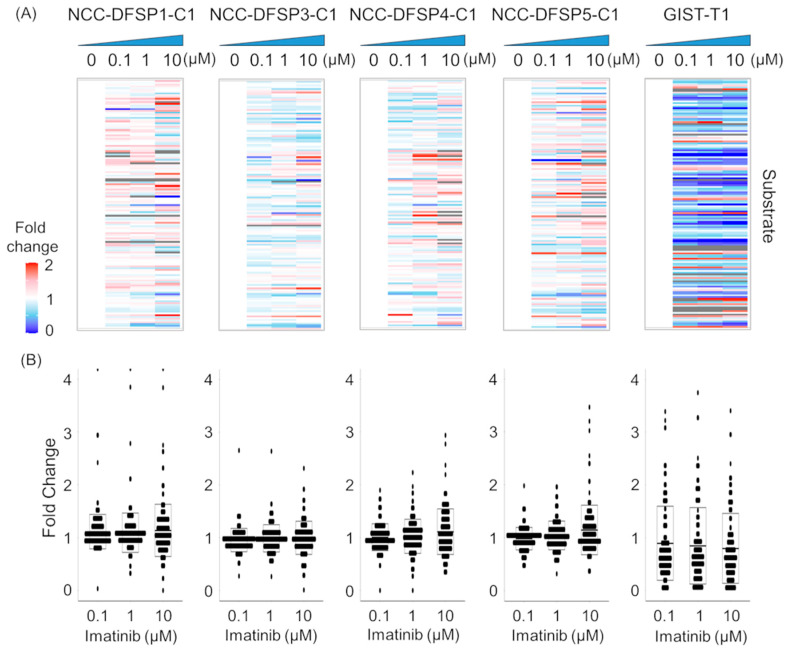
Effect of imatinib on kinase activity of the DFSP PDCs and GIST-T1 cells (**A**) The heatmap was generated by inferring kinase activity against fold change (each concentration/DMSO). Fold change in kinase activity with imatinib concentration. Lysates were incubated with 0, 0.1, 1, or 10 µM imatinib (n = 2). Highly activated indicated red, and poorly activated blue. (**B**) Scatter plots of the change in fold change (signal intensity with imatinib/signal intensity without imatinib) for each substrate from the in vitro kinase array in the five cell lines).

**Figure 3 cells-14-00884-f003:**
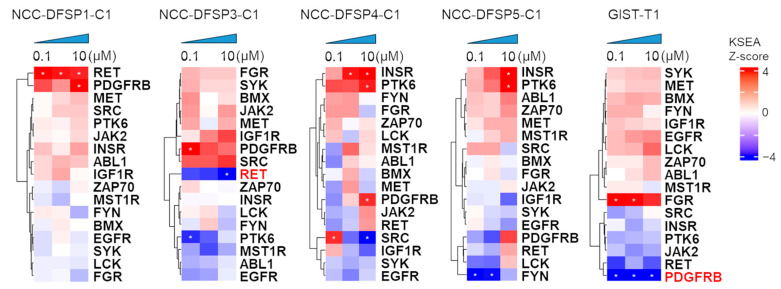
Kinase substrate enrichment analysis for imatinib. Heatmap showing the distribution of KSEA Scores among the four DFSP PDCs and GIST-T1 cells. Kinase activities of cells treated with 0, 0.1, 1, and 10 µM imatinib were measured. Significant changes are indicated by asterisks. Red characters show names of kinases with significant inhibition by imatinib. High KSEA scores denote activation and low scores denote inhibition.

**Figure 4 cells-14-00884-f004:**
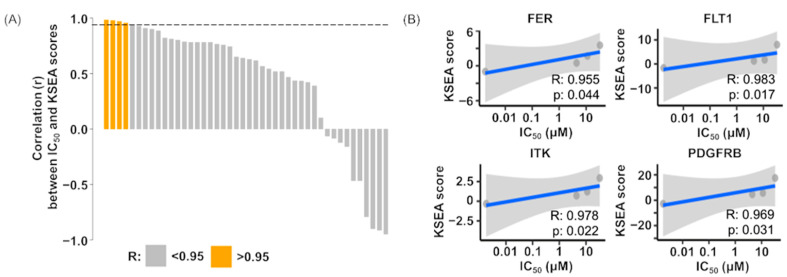
Correlation analysis of drug sensitivity on imatinib and kinase activity. (**A**) Waterfall plot summarizing the Spearman correlation between drug sensitivity and kinase activity in 4 DFSP PDCs and GIST-T1 cell lines. Spearman correlation coefficients (r values) were calculated between the KSEA scores of individual kinases and imatinib IC_50_ values for the five cell lines. The x-axis represents kinases, and the y-axis shows the cumulative sum of Spearman r values for 44 kinases. Orange bars indicate kinases with a strong correlation (r > 0.95). (**B**) Scatter plot showing the Spearman correlation between the KSEA scores and imatinib IC_50_ values across the five cell lines. Each dot corresponds to one cell line.

**Table 1 cells-14-00884-t001:** Overview of patient characteristics in this study.

Cell Line Name	Diagnosis *	Age	Gender	Location	Site of Origin	Reference
NCC-DFSP1-C1	FS-DFSP	46	M	Primary	Abdominal wall	[15]
NCC-DFSP3-C1	FS-DFSP	51	F	Metastatic	Left thigh	[16]
NCC-DFSP4-C1	FS-DFSP	60	M	Primary	Left lower back	[17]
NCC-DFSP5-C1	DFSP	52	M	Primary	Left proximal tibia	[18]

* DFSP: dermatofibrosarcoma protuberans; FS-DFSP: fibrosarcomatous dermatofibrosarcoma protuberans.

**Table 2 cells-14-00884-t002:** Overview of drug sensitivity profiles in ten candidate kinase inhibitors.

	Drug Name	NCC-DFSP1-C1	NCC-DFSP3-C1	NCC- DFSP4-C1	NCC-DFSP5-C1	GIST-T1
10 μM cell viability (%)	Imatinib	76.76	41.42	67.00	92.11	13.24
	Imatinib Mesylate	85.85	41.11	77.40	93.28	10.14
	Cediranib	2.35	2.81	14.23	7.92	107.79
	Crizotinib	4.05	12.56	0.60	2.66	111.10
	Ceritinib	13.04	26.92	0.54	0.00	80.77
	Entrectinib	51.03	82.41	4.55	0.06	19.36
	Sunitinib Malate	32.96	46.95	5.68	11.92	8.53
	Foretinib	15.13	16.56	20.55	0.01	−1.20
	Osimertinib	18.77	1.89	1.21	0.68	−0.04
	Ponatinib	0.34	0.96	0.40	1.34	−1.49
IC_50_ (μM)	Imatinib	32.6900	4.4190	11.1700	11.7800	0.0018
	Imatinib Mesylate	42.0700	2.3960	13.1900	4.5540	0.0136
	Cediranib	0.0050	0.8100	1.3500	4.6200	0.0035
	Crizotinib	0.0690	1.2300	2.1900	1.2500	0.2406
	Ceritinib	0.7900	3.9100	1.8700	2.2600	2.3630
	Entrectinib	0.0160	2.0600	1.5600	2.0300	1.3510
	Sunitinib Malate	4.6800	9.4100	6.5200	7.3000	0.0006
	Foretinib	0.2600	0.5300	0.6300	1.1800	0.6235
	Osimertinib	0.1500	2.6200	0.5700	2.0600	0.9135
	Ponatinib	0.0500	0.0800	0.3600	1.3700	0.0883

**Table 3 cells-14-00884-t003:** Gene ontology analysis of kinase activity profiles treated with imatinib in four DFSP PDCs and GIST-T1 cells.

Sample	Term Description	NES	Exact *p*-Value	False Discovery Rate
NCC-DFSP1-C1	MAPK signaling pathway	1.5	0.003	0.022
NCC-DFSP3-C1	EGFR tyrosine kinase inhibitor resistance	1.811	0.001	0.005
NCC-DFSP3-C1	MAPK signaling pathway	−1.362	0.012	0.024
NCC-DFSP4-C1	NA	NA	NA	NA
NCC-DFSP5-C1	NA	NA	NA	NA
GIST-T1	MAPK signaling pathway	−1.747	0.001	0.008

## Data Availability

Data is contained within the article and Appendix A.

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
