# Peer review of "Pharmacokinomic Profiling Using Patient-Derived Cell Lines Predicts Sensitivity to Imatinib in Dermatofibrosarcoma Protuberans"

_cells, 2025, doi:10.3390/cells14120884_

Round 1

Reviewer 1 Report

Comments and Suggestions for Authors

In the present work, Noguchi et al. investigated the pharmacokinomic profile of tyrosine kinase inhibitors in patient-derived cell lines of dermatofibrosarcoma protuberans. The present work is very interesting and has merit for publication after addressing some minor issues.

The authors have performed a great deal of experimental work, which make their work very interesting to read and I found it very informative.

The authors discuss the effects of imatinib and imatinib mesylate with respect to cell viability, yet five other inhibitors appear to be more effective (in particular, sunitinib, foretinib, osimertinib and ponatinib). Why is that difference observed? In particular, these inhibitors appear to be effective in all cell lines. Why not investigate those in particular? How is to explain that these inhibitors are effective at all different cell types and the others tested are not? What mechanisms are expected to influence such an effect? Please comment on that.

Why did the authors focused on imatinib and not those inhibitors that appeared to be the most effective with respect to cell proliferation and viability inhibition?

Finally, please highlight your findings. How are these findings to help tumor prognosis and therapy? How is it possible to unravel tumor mechanics from in vitro systems? Please comment on that.

Please move Table 1 near to the first reference to Table 1.

Overall, the present work is well-written and concise. It has merit for publication.

Author Response

Reviewer 1

Comment:

In the present work, Noguchi et al. investigated the pharmacokinomic profile of tyrosine kinase inhibitors in patient-derived cell lines of dermatofibrosarcoma protuberans. The present work is very interesting and has merit for publication after addressing some minor issues.

The authors have performed a great deal of experimental work, which make their work very interesting to read and I found it very informative.

Response to comment:

We appreciate the reviewer’s comment and suggestion to improve our manuscript. One-by-one responses have been provided as follows:

Comment 1:

The authors discuss the effects of imatinib and imatinib mesylate with respect to cell viability, yet five other inhibitors appear to be more effective (in particular, sunitinib, foretinib, osimertinib and ponatinib). Why is that difference observed? In particular, these inhibitors appear to be effective in all cell lines. Why not investigate those in particular? How is to explain that these inhibitors are effective at all different cell types and the others tested are not? What mechanisms are expected to influence such an effect? Please comment on that.

Response to comment 1:

We appreciate the reviewer’s insightful question regarding the observed differences in efficacy between imatinib and other inhibitors such as sunitinib, foretinib, osimertinib, and ponatinib. As noted, several of these compounds appeared more effective at a single 10 μM screening concentration across all tested cell lines. However, it is well known that single-concentration screening does not always correlate precisely with ICâ‚…â‚€ values. Some compounds may show modest inhibition at 10 μM, but exhibit steep dose-response curves with low ICâ‚…â‚€ values (Gubler et al., 2012; Inglese et al., 2006). Conversely, compounds that appear effective at 10 μM may not maintain selectivity or potency across concentrations. In our study, we focused further ICâ‚…â‚€ analyses only on compounds that showed both strong inhibition at 10 μM and selective activity in certain cell lines. The compounds highlighted by the reviewer, although potent at 10 μM, demonstrated universally low ICâ‚…â‚€ values across all lines, suggesting possible nonspecific cytotoxicity or broad target activity. Therefore, we did not prioritize them for further mechanistic investigation at this stage. Nonetheless, we agree that their broad efficacy warrants future study, particularly to dissect whether the observed pan-activity is driven by common molecular vulnerabilities in these sarcoma cell lines.

References:

  • Gubler H, et al. J Biomol Screen. 2013 Jan;18(1):1-13. PMID: 22853931
  • Inglese J, et al. Proc Natl Acad Sci U S A. 2006;103(31):11473–8. PMID: 16864780

Comment 2:

Why did the authors focused on imatinib and not those inhibitors that appeared to be the most effective with respect to cell proliferation and viability inhibition?

Response to comment 2:

We really thank appreciate the reviewer’s insightful comment. As noted, several inhibitors other except imatinib did show promising effects on cell proliferation and viability. However, we could not identify any single inhibitor that exhibited consistently low IC50 values across all DFSP PDCs. Therefore, we chose to focus on imatinib primarily because it is already an approved treatment for DFSP. Our aim was to explore its potential further in the context of our study, while acknowledging that other inhibitors may also warrant investigation in future research.

Comment 3:

Finally, please highlight your findings. How are these findings to help tumor prognosis and therapy? How is it possible to unravel tumor mechanics from in vitro systems? Please comment on that.

Response to comment 3:

We thank the reviewer’s insightful question. Our findings suggest that patient-derived sarcoma cell lines with higher PDGFR pathway activity tend to be more sensitive to imatinib, which may translate into a better clinical prognosis. Conversely, cell lines with low PDGFR activity showed resistance, potentially reflecting poorer outcomes. These results indicate that in vitro models, when molecularly characterized, can mirror treatment response and biological aggressiveness observed in patients. By incorporating additional well-annotated clinical samples including long-term survivors, we believe it will be possible to construct a robust prognostic framework. Ultimately, such systems may not only guide therapeutic decisions but also help unravel the molecular underpinnings of tumor progression and resistance mechanisms.

Comment 4:

Please move Table 1 near to the first reference to Table 1.

Response to comment 4:

We would like to thank the reviewer for this helpful suggestion. In accordance with the comment, we have moved Table 1 to appear near the first reference to it in the manuscript to improve clarity and readability. Table 1 is now located on page 3.

Comment 5:

Overall, the present work is well-written and concise. It has merit for publication.

Response to comment 5:

We sincerely thank the reviewer for the positive feedback and appreciation of our work. We are encouraged by your comments and will continue to refine our research further.

Reviewer 2 Report

Comments and Suggestions for Authors

This study presents a comprehensive pharmacokinomic investigation using patient-derived cell lines (PDCs) of dermatofibrosarcoma protuberans (DFSP), aiming to identify biomarkers predictive of imatinib sensitivity. The integration of kinase activity profiling (PamChip) with drug screening across 60 tyrosine kinase inhibitors provides a novel strategy to anticipate drug resistance and guide therapy. The authors identify FER, ITK, VEGFR1 (FLT1), and PDGFRB as kinases strongly correlated with imatinib responsiveness.

DFSP is a rare sarcoma with variable sensitivity to imatinib. This work represents an important step toward precision medicine for DFSP.

The integration of drug screening, kinase activity assays, and correlation analyses is methodologically sound and well-executed.

The use of four newly established DFSP cell lines is a major strength, considering the scarcity of such models.

However, while the study identifies FER, FLT1, ITK, and PDGFRB as predictive markers, the findings are based exclusively on in vitro data without validation in clinical samples or in vivo models. Add a statement in the discussion acknowledging this limitation and suggesting future validation using patient tumor specimens or animal models.

The mechanisms underlying resistance remain speculative, particularly regarding differential phosphorylation patterns.

Abstract: Clearly state that results are based on in vitro assays and patient-derived cells only.

Ensure all figures and supplementary tables are appropriately referenced and included in the submission.

The IC50 values for imatinib in DFSP lines vary widely (4–32 µM). Clarify if these are within pharmacologically achievable concentrations.

Statistical Analysis: FDR correction is appropriately used. Consider reporting exact p-values in addition to significance thresholds.

Author Response

Reviewer 2

Comment:

This study presents a comprehensive pharmacokinomic investigation using patient-derived cell lines (PDCs) of dermatofibrosarcoma protuberans (DFSP), aiming to identify biomarkers predictive of imatinib sensitivity. The integration of kinase activity profiling (PamChip) with drug screening across 60 tyrosine kinase inhibitors provides a novel strategy to anticipate drug resistance and guide therapy. The authors identify FER, ITK, VEGFR1 (FLT1), and PDGFRB as kinases strongly correlated with imatinib responsiveness. DFSP is a rare sarcoma with variable sensitivity to imatinib. This work represents an important step toward precision medicine for DFSP. The integration of drug screening, kinase activity assays, and correlation analyses is methodologically sound and well-executed. The use of four newly established DFSP cell lines is a major strength, considering the scarcity of such models.

Response to comment:

The reviewer’s comment is highly appreciated. We have provided one-by-one responses to the comments and concerns as follows:

Comment 1:

However, while the study identifies FER, FLT1, ITK, and PDGFRB as predictive markers, the findings are based exclusively on in vitro data without validation in clinical samples or in vivo models. Add a statement in the discussion acknowledging this limitation and suggesting future validation using patient tumor specimens or animal models.

Response to comment 1:

Accordingly, we have added a statement for the limitation in 4.4 Limitations of the discussion as follows: “Fourth, our findings are derived solely from in vitro data and lack validation in clinical samples or in vivo models. To confirm the clinical relevance of the identified biomarkers, further studies using patient-derived tumor samples or animal models will be required. Given the rarity of DFSP, such validation efforts will likely necessitate multiple institutional collaboration.”

Comment 2:

The mechanisms underlying resistance remain speculative, particularly regarding differential phosphorylation patterns.

Response to comment 2:

We agree the reviewer’s comment and have addressed this point in section 4.4 “Limitations” of the discussion by adding the following: “Fifth, our findings of the mechanisms underlying resistance, based on in vitro kinase activity assay, remain speculative. To validate these hypotheses, additional functional studies including overexpression and/or knockdown experiments are necessary to strengthen the evidence. We plan to pursue such experiments in future research.”

Comment 3:

Abstract: Clearly state that results are based on in vitro assays and patient-derived cells only.

Response to comment 3:

Accordingly, we have revised abstract emphasizing that our results are based on in vitro assays and patient-derived cells only as follows:

Abstract: Dermatofibrosarcoma protuberans (DFSP) is a rare sarcoma, characterized by a COL1A1-PDGFB fusion. Imatinib, a multi-target tyrosine kinase inhibitor, is a standard treatment of DFSP. However, resistance emerges in 10-50% of cases. There is an urgent need for predictive biomarkers to refine the patient selection and improve therapeutic outcomes. We aimed to identify predictive biomarkers for imatinib response and explored a pharmacokinomic approach using in vitro assays with patient-derived DFSP cell lines. Four DFSP cell lines we established were analyzed for tyrosine kinase activities on PamChip, three-dimensional peptide array in the presence and absence of imatinib, along with an imatinib-sensitive cell line, GIST-T1, as a positive control. Drug screening was also performed using 60 FDA-approved tyrosine kinase inhibitors, including imatinib. The kinomic profiles were compared with the kinase inhibitor screening results to identify predictive druggable targets. Drug sensitivity was associated with increased activity of PDGFRB, as indicated by the PamChip assay and Western blotting. Furthermore, imatinib sensitivity correlated to the activity of three kinases: FER, ITK, and VEGFR1, suggesting their potential as potential predictive biomarkers. Our cell-based pharmacokinomic approach using patient-derived DFSP cell lines would facilitate the identification of resistant cases to imatinib and guide alternative therapeutic strategies.

Comment 4:

Ensure all figures and supplementary tables are appropriately referenced and included in the submission.

Response to comment 4:

We are thankful for the reviewer’s careful attention to the consistency and completeness of our manuscript. In response to the comment, we have made the following revisions:

  • The label "Figure 1B" has been corrected to "Figure 1" to maintain consistent figure numbering.
  • We noticed that Figure 4 was missing in the previous version. It has now been appropriately inserted and referenced in the main text.
  • We have double-checked all figure and supplementary table references to ensure that they are accurately cited and included in the revised submission.

Thank you for pointing this out and helping us improve the clarity and completeness of the manuscript.

Comment 5:

The IC50 values for imatinib in DFSP lines vary widely (4–32 µM). Clarify if these are within pharmacologically achievable concentrations.

Response to comment 5:

We thank the reviewer’s insightful comment. Clinically, the Cmax of imatinib in patients ranges between approximately 3 to 8 µM depending on the dosage (400–800 mg/day) [Larson et al., 2008 (PMID: 18256322)]. In our study, DFSP lines showed IC50 values ranging from 4 to 32 µM. While some of the cell lines exhibit IC50 values within the pharmacologically achievable range, others exceed it. This variation suggests potential differences in intrinsic or acquired resistance mechanisms among DFSP lines and highlights the need for further investigation into biomarkers that predict imatinib sensitivity.

Comment 6:

Statistical Analysis: FDR correction is appropriately used. Consider reporting exact p-values in addition to significance thresholds.

Response to comment 6:

Thank you for your suggestion. To address this point, we revised the description in Section 2.8 to clarify that both raw p-values and FDR-adjusted p-values were calculated and reported in Table 3.

Reviewer 3 Report

Comments and Suggestions for Authors

The problem of second line therapy in metastatic soft tissue sarcoma is widely diffused in most histotypes  and represents a great challenge  for the clinicians.

Primary and secondary resistance to therapy are   a well known barrier to the complete cure of the disease.  In many STS the time to progression does not exceed 4 months. As  consequence  OS is about 10 months.

DFSP  is not an exception and when this tumor becomes  metastatic or inoperable the chance of cure is  reduced.

Since 2004 Imatinib  demonstrated some activities in DFSP but the benefit is modest.

Either palliative Radiotherapy or   Chemotherapy can offer a short lasting benefit.

 The study of Noguchi et Al offers a new perspective:  in vitro drug screening with 60 different TKI in order to define the activity of every single molecule  on DFSP cell lines.

The study is well conceived,  The cell culture procedure and  drug screening are well described.  The figures are clear and representative of the experiments. The limitations of the  study are  well   reported and discussed . In conclusion   the identification of  some predictive biomarkers in DFSP could open a new interesting frontier in the treatment of this rare disease.

Moreover this experience  could be transferred in other  neoplasm treated with TKI ( GIST, renal cancer) to predict the response to therapy.

The little number of cases (4)  is the unique  important limit of the study 

Author Response

Comment:

The problem of second line therapy in metastatic soft tissue sarcoma is widely diffused in most histotypes  and represents a great challenge for the clinicians.

Primary and secondary resistance to therapy are a well known barrier to the complete cure of the disease.  In many STS the time to progression does not exceed 4 months. As consequence OS is about 10 months.

DFSP is not an exception and when this tumor becomes metastatic or inoperable the chance of cure is reduced.

Since 2004 Imatinib demonstrated some activities in DFSP but the benefit is modest.

Either palliative Radiotherapy or Chemotherapy can offer a short lasting benefit.

 The study of Noguchi et Al offers a new perspective:  in vitro drug screening with 60 different TKI in order to define the activity of every single molecule on DFSP cell lines.

The study is well conceived, The cell culture procedure and drug screening are well described.  The figures are clear and representative of the experiments. The limitations of the study are well   reported and discussed. In conclusion   the identification of some predictive biomarkers in DFSP could open a new interesting frontier in the treatment of this rare disease.

Moreover this experience could be transferred in other neoplasm treated with TKI ( GIST, renal cancer) to predict the response to therapy.

The little number of cases (4) is the unique important limit of the study 

Response to comment:

We sincerely thank the reviewer for their thoughtful and encouraging comments. We appreciate the recognition of the rationale and design of our study, as well as the clarity of the figures and the discussion of the study's limitations. As the reviewer noted, the limited number of DFSP cases (n=4) is a key limitation. This point has been acknowledged and discussed in the manuscript. We hope that our findings will lay the groundwork for future studies with larger cohorts to validate the results and further investigate predictive biomarkers in DFSP and other TKI-treated neoplasms.